# Popcorn Transitions and Approach to Conformality in Homogeneous Holographic Nuclear Matter

Jesús Cruz Rojas [1] , Tuna Demircik [2] and Matti Järvinen [1,3,*]

1 Asia Pacific Center for Theoretical Physics, Pohang 37673, Republic of Korea
2 Institute for Theoretical Physics, Wrocław University of Science and Technology, 50-370 Wrocław, Poland
3 Department of Physics, Pohang University of Science and Technology, Pohang 37673, Republic of Korea
* Correspondence: matti.jarvinen@apctp.org

**Abstract:** We study cold and dense nuclear matter by using the gauge/gravity duality. To this end, we use the Witten–Sakai–Sugimoto model and the V-QCD models with an approach where the nuclear matter is taken to be spatially homogeneous. We focus on the "popcorn" transitions, which are phase transitions in the nuclear matter phases induced by changes in the layer structure of the configuration on the gravity side. We demonstrate that the equation of state for the homogeneous nuclear matter becomes approximately conformal at high densities, and compare our results to other approaches.

**Keywords:** gauge/gravity duality; quantum chromodynamics; dense matter

## 1. Introduction

Recent observations of neutron star mergers by the LIGO/Virgo collaboration have opened a new window for studying dense matter in quantum chromodynamics (QCD). In particular, the gravitational and electromagnetic waves observed from the GW170817 merger event [1] have already set highly nontrivial constrains for the QCD equation of state [2] at low temperatures and high densities. This progress has boosted interest in theoretical studies of dense QCD, which is a challenging topic, as standard theoretical and computational tools do not work in extensive regions of the phase diagram (see the overview in [3]). These include the region of dense nuclear matter, i.e., a nucleon liquid at baryon number densities well above the nuclear saturation density $\rho_s \approx 0.16\,\mathrm{fm}^{-3}$.

The difficulty of solving the properties of dense matter calls for new methods. A possibility is to use the gauge/gravity duality. Indeed, applications of holographic QCD models to dense matter have received significant interest recently. There has been progress in developing models for quark matter [4–9], nuclear matter [10–14], and other phases (such as color superconducting or quarkyonic phases) [15–18]. See also the reviews [19,20].

The natural starting point for describing nuclear matter is to study the holographic duals for nucleons. The standard approach [21] boils down to describing them as solitonic "instanton" solutions of bulk gauge fields, i.e., the gauge fields living in the higher dimensional gravity theory. These solitons are localized both in the spatial directions and in the holographic direction (but not in time). Solitons that are duals of isolated nucleons have been solved in various holographic models [22–31]. However, constructing more complicated solutions and eventually the holographic dual of dense nucleon matter out of these bulk solitons is challenging. Some results, which use instanton gases without interactions, are available [13,32–36] and also include two-body interactions [10]. Moreover, at large $N_c$, the nuclear matter is a crystal rather than a liquid of nucleons [37]. Such crystals have been studied by using different toy models and approximations [38–43].

In this article, we focus on a simpler approach, which treats dense nuclear matter as a homogeneous configuration of the non-Abelian gauge fields in the bulk. This approach was applied to the Witten–Sakai–Sugimoto (WSS) model [44–46] in [47] and argued to be a reasonable approximation at high density. An even simpler approach is to treat the

baryons as point-like sources in the bulk, which may be a better approximation at low density [47,48]. The homogeneous approach was further developed in [49,50] and applied to other models in [11,14]. Interestingly, dense (and cold) homogeneous holographic nuclear matter was seen to have a high speed of sound, clearly above the value $c_s^2 = 1/3$ of conformal theories [11] (see also [51,52]). That is, the equation of state is "stiff". This is important, as it helps to construct models that pass observational bounds [53–55].

Changes in the structure of dense nuclear matter may give rise to transitions within the nuclear matter phase. At large $N_c$, nuclear matter is a crystal of skyrmions, solitons of the low energy chiral effective theory [56]. As the density increases, the skyrmion crystal is expected to undergo a transition into half-solitons, where each node of the crystal carries baryon number of one half [57–60]. Similar structures have been studied by using the gauge/gravity duality in [39].

This topology changing transition has been studied extensively by using an effective field theory approach, which introduces the $\sigma$ meson of QCD as a pseudo-Nambu–Goldstone mode of broken scale invariance, and vector mesons through the hidden local symmetry approach [61,62]. This approach is supported by the analysis of the nucleon axial coupling $g_A$ for heavy nuclei [63]. Above the transition density, it was found that the speed of sound rapidly approaches the conformal value $c_s^2 = 1/3$ [64]. At the same time, the polytropic index $\gamma = d \log p / d \log \epsilon$ takes small values [65,66] compared to what is usually found in nuclear theory models [67,68]. The transition has also been argued [69] to be indicative of quark–hadron continuity [70], which states that there is no phase transition between nuclear and quark matter. Whether the continuity is a feasible possibility is a matter of ongoing debate' see [71–74] (also, [15] for a holographic discussion).

A closely related transition realized in holographic setups is the transition from a single-layer configuration into a double layer configuration. Recall that in the holographic method, each nucleon is dual to a five dimensional soliton on the gravity side, and the dual nuclear matter is therefore obtained as an ensemble of such solitons. In the low density limit, the location of each soliton is found by individually minimizing its energy, so that the solitons form a single layer at a specific value of the holographic coordinate. For dense configurations, however, the repulsive interactions between solitons will eventually force them out of this layer, which leads to a double layer or a more complicated configuration. This transition was coined the "popcorn" transition in [40]. If interactions between the solitons are attractive at large distances (as is the case for real QCD), the picture is more complicated, as the solitons clump together even at low densities, but the transition may still be present. Various phases appear as the density increases further [41,42] in setups motivated by the WSS model. The simplest case of the transition is, however, the separation of a single layer into two layers. This kind of transition was also found to take place in the WSS model in various approximations: when the instantons were approximated as point-like objects [17], when including finite widths [36], and when using a homogeneous approach [50]. Indications of such a transition were also seen when using a homogeneous Ansatz for nuclear matter in the hard wall model of [14], where it was interpreted as a transition to a quarkyonic phase [75].

In this article, we study the popcorn transitions within cold homogeneous holographic nuclear matter by using two different models: the top-down WSS model and the bottom-up V-QCD model [19,76]. These two are arguably the most developed holographic top-down and bottom-up models for QCD at finite temperature and density. For the WSS model, a similar analysis was carried out in [50]. This reference used an approach which is slightly different from ours; in their case, a zero curvature condition for the non-Abelian gauge fields in the Lagrangian density is imposed before approximating the density to be homogeneous. We use a somewhat simpler approach where the fields are assumed to be homogeneous to start with. In our case, as we will discuss in detail below, a discontinuity of the gauge fields as a function of the holographic coordinate is required to have nonzero baryon density [47]. This may appear to be a weakness of the simpler approach, but we remark that the discontinuity is

actually well motivated, as it can be seen to arise from the non-analyticity of the instanton solutions at their centers after smearing over the spatial dimensions [19].

The main goal in this article is to analyze the softening of the equation of state at the phase transition. The main indicators for this are the speed of sound and the polytropic index $\gamma$. We compute these quantities in both holographic models and compare them to results in other setups. In particular, we find interesting similarities with the effective theory approach for the topology changing transition [62,64].

The rest of the article is organized as follows. In Section 2, we review the setup with homogeneous nuclear matter for the WSS model; in Section 3, we do the same for the V-QCD model. In Section 4, we discuss the numerical results for the solutions, the phase transitions, and the equation of state. Finally, we discuss our findings in Section 5.

## 2. Homogeneous Nuclear Matter in the Witten–Sakai–Sugimoto Model

The phase diagram of QCD has been studied by using several holographic "top-down" models, i.e., models directly based on string theory, such as the Witten–Sakai–Sugimoto model [45,46]. In this model, Witten's non-supersymmetric model for low-energy QCD [44] has been successfully applied to study the spectra and the properties of mesons and baryons.

In the WSS model, the pure glue physics of the QFT is described by the dual gravitational background and is sourced by $N_c$ D4-branes in type-IIA superstring theory. Fundamental degrees of freedom are included by adding $N_f$ pairs of D8 and $\overline{D8}$-branes, such that the strings connecting $D4 - D8$ and $D4 - \overline{D8}$ branes are associated with left- and right-handed fermions.

Witten's model includes a phase transition involving a topologically nontrivial change in geometry from a low temperature "cigar" geometry to a high temperature black hole geometry [77]. We focus here on the low temperature geometry, which we will give explicitly below. In the low temperature geometry, the D8 and $\overline{D8}$-branes join at the tip of the cigar (see Figure 1), which locks together the flavor transformations on the branes, indicating chiral symmetry breaking. As shown in the figure, we assume the simplest case where the D8 and $\overline{D8}$-branes are antipodal, i.e., are located at exactly opposite curves on the cigar. In this setting, all quark masses are equal to zero. A chemical potential for the baryon number can be turned on by adding a nonzero source for the temporal component of the Abelian gauge field on the D8-branes.

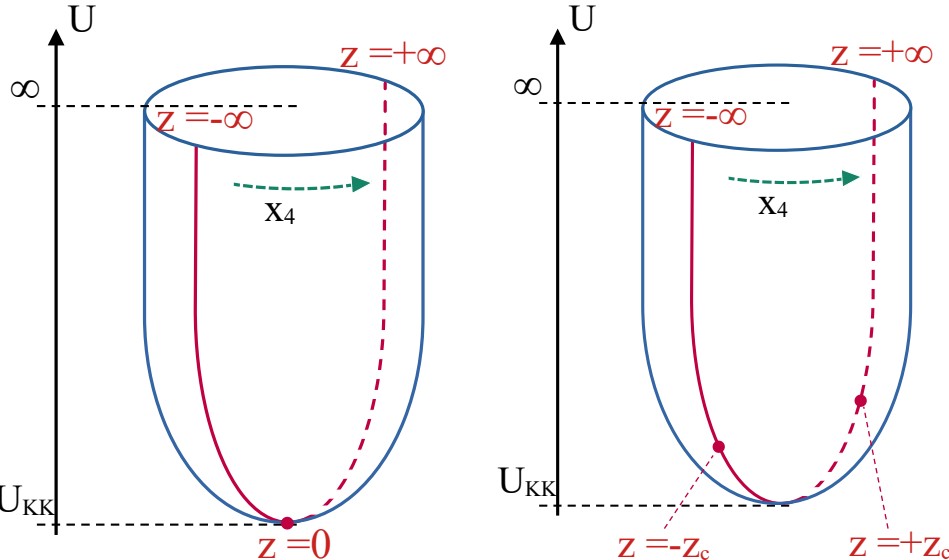

**Figure 1.** Setup in the WSS model. The coordinate $z$ runs between $z = -\infty$ and $z = \infty$ between the two boundaries of the D8 brane embedding as indicated in the figure. The blobs show the locations of the discontinuities for the single-layer configuration (**left**) and for the double-layer configuration (**right**).

### 2.1. Expanding the Dirac-Born-Infeld Action

The 10-dimensional metric of the confined low temperature geometry in the Witten model can be written as [78,79]

$$ds^2 = \left(\frac{U}{R}\right)^{3/2}\left[dx_\mu dx^\mu + f(U)dx_4^2\right] + \left(\frac{R}{U}\right)^{3/2}\left[\frac{dU^2}{f(U)} + U^2 d\Omega_4^2\right] \tag{1}$$

where $R$ is the curvature radius,

$$f(U) = 1 - \frac{U_{KK}^3}{U^3} \tag{2}$$

with $U_{KK}$ denoting the end of space and $d\Omega_4^2$ the metric of $S^4$. For the Minkowski metric $dx_\mu dx^\mu$, we use mostly plus conventions, and the $x_4$ coordinate is compactified on a circle. The dilaton is given by

$$e^\phi = g_s \left(\frac{U}{R}\right)^{3/4}, \tag{3}$$

where $g_s$ is the string coupling.

In the $(x_4, U)$-coordinates, this geometry takes the form of a cigar and the regularity at the tip of the cigar links the radius of compactification $R_4$ of the $x_4$ coordinate to the Kaluza–Klein scale characterized by $U_{KK}$, as $R_4 = \frac{(4\pi)R^{3/2}}{3\sqrt{U_{KK}}}$; we can also define the mass scale $M_{KK}^{-1} = \frac{R_4}{2\pi}$. The simplest D8 brane embedding within the cigar geometry is the antipodal one, given (for example) by $x_4 = 0$ and $x_4 = \pi R_4$. By changing the coordinates to $U = U_{KK}(1 + z^2)^{1/3}$ the induced metric on the brane can be written as

$$ds_{\rm ind}^2 = \left(\frac{U_{KK}}{R}\right)^{3/2}\sqrt{1 + z^2}\,dx_\mu dx^\mu + \left(\frac{R}{U_{KK}}\right)^{3/2}\frac{4dz^2}{9(1 + z^2)^{5/6}} + R^{3/2}\sqrt{U_{KK}}\sqrt[6]{1 + z^2}d\Omega_4^2\,. \tag{4}$$

Here, the coordinate $z$ takes both positive and negative values on different branches of the brane. The boundary is at $z = \pm\infty$ and the tip of the cigar at $z = 0$. See Figure 1 for illustration.

We work in units where $U_{KK} = 1$ and $R^3 = 9/4$, making all quantities dimensionless. This can be interpreted such that we show dimensionful quantities in units of the Kaluza–Klein mass $M_{KK}$, which is also set to one. We also start from the Dirac–Born–Infeld action

$$S_{\rm DBI} = -\tau_8 \int d^9 x\, e^{-\phi}{\rm tr}\sqrt{-\det(g + \mathcal{F})}\,, \tag{5}$$

where the trace is over flavor indices. The brane tension is given by

$$\tau_8 = \frac{1}{(2\pi)^8 l_s^9} = \frac{\lambda^{9/2}}{157464\sqrt{2}\pi^8}\,. \tag{6}$$

Here, $l_s$ is the string length, $\lambda$ is the 't Hooft coupling, and $\mathcal{F}$ is the field strength tensor of the gauge field $\mathcal{A}$. We used the relations $R^3 = \pi g_s N_c l_s^3$ and $2\pi l_s g_s N_c = \lambda$.

We use a similar expansion as in the case of V-QCD below [11] so that the non-Abelian components of the gauge fields are treated as small, but the Abelian terms are kept unexpanded. To do so, we separate the gauge field into non-Abelian and Abelian components: $\mathcal{A} = A + \hat{A}$, where $A$ is non-Abelian and $\hat{A}$ is Abelian, i.e., proportional to the unit matrix in flavor space (and similarly, $\mathcal{F} = F + \hat{F}$ for the field strengths). We take only the temporal component of the Abelian gauge field to be nonzero, assume that it depends only on the holographic coordinate $z$, and assume no dependence on the angular coordinates of $\Omega_4$ for all fields, so that these coordinates can be integrated out. Then, the

five-dimensional effective action for the gauge fields to leading nontrivial order in the non-Abelian field strength tensor $F$ is given as

$$S = S_{\text{DBI}}^{(0)} + S_{\text{DBI}}^{(1)} + S_{\text{CS}} \tag{7}$$

where the terms arising from the Dirac–Born–Infeld action read

$$S_{\text{DBI}}^{(0)} = -\frac{\lambda^3 N_c N_f}{19683\pi^5} \int d^5x \sqrt[3]{1+z^2} \sqrt{(1+z^2)^{2/3} - (1+z^2)\Phi'(z)^2} \tag{8}$$

and

$$
S_{\text{DBI}}^{(1)} = -\frac{\lambda N_c}{216\pi^5} \int d^5x \ \text{tr}\left[ -\frac{F_{tz}^2\left(1+z^2\right)}{\left(1 - \sqrt[3]{1+z^2}\Phi'(z)^2\right)^{3/2}} - \frac{F_{ti}^2}{\sqrt[3]{1+z^2}\sqrt{1 - \sqrt[3]{1+z^2}\Phi'(z)^2}} \right.
$$
$$
\left. + \frac{F_{ij}^2\sqrt{1 - \sqrt[3]{1+z^2}\Phi'(z)^2}}{2\sqrt[3]{1+z^2}} + \frac{F_{zi}^2\left(1+z^2\right)}{\sqrt{1 - \sqrt[3]{1+z^2}\Phi'(z)^2}} \right], \tag{9}
$$

where the spatial indices $i, j$ are summed over. Notice that the general Dirac–Born–Infeld action is ambiguous for non-Abelian fields, but up to second order in the expansion the action is non-ambiguous. The Chern–Simons term is

$$S_{\text{CS}} = \frac{N_c}{24\pi^2} \int \left\{ \omega_5 + d\left[ \hat{A} \wedge \text{tr}\left( 2A \wedge F + \frac{i}{2}A \wedge A \wedge A \right) \right] + 3\hat{A} \wedge \text{tr}(F \wedge F) \right\} \tag{10}$$

with the Abelian gauge field normalized as $\Phi = 2\lambda \hat{A}_t / (\sqrt{729}\pi)$. Here

$$\omega_5 = \text{tr}\left( A \wedge F \wedge F + \frac{i}{2}A \wedge A \wedge A \wedge F - \frac{1}{10}A \wedge A \wedge A \wedge A \wedge A \right) \tag{11}$$

gives the standard Chern0-Simons term for the brane. We used conventions where $F = dA - iA \wedge A$. Notice that in (10), the Abelian field couples to the instanton density in the bulk as expected (see the last term). Indeed, notice that $S_{DBI}^{(0)}$ and $S_{DBI}^{(1)}$ depend on the Abelian gauge field only through its $z$-derivative, and only $S_{\text{CS}}$ contains non-derivative dependence on this field. Since the total baryon charge density is defined as

$$\rho_0 = -\left( \frac{\delta S}{\delta \hat{A}_t'} \right)_{\text{bdry}} = \int dz \frac{\delta S}{\delta \hat{A}_t}, \tag{12}$$

according to the holographic dictionary, the baryon charge is given by the coupling of the non-Abelian field to $\hat{A}_t$ in $S_{\text{CS}}$. In other words, the Chern–Simons term determines how the solitons source baryonic charge.

    We also remark that the construction of the precisely consistent Chern–Simons term is actually rather involved in general [80], but in the simple case considered here, complications do not arise.

### 2.2. The Homogeneous Ansatz

    Then, as the next step, we set $N_f = 2$ and insert the homogeneous Ansatz

$$A^i = h(z)\sigma^i \tag{13}$$

where $h(z)$ is a scalar function and $\sigma^i$ are the Pauli matrices. The non-Abelian $A_t$ and $A_z$ components are set to zero. We then find that

$$F_{zi} = h'(z)\sigma_i \,, \qquad F_{ij} = 2h(z)^2 \epsilon_{ijk}\sigma^k \tag{14}$$

while other components of the field strength are zero. We obtain

$$S_{\text{DBI}}^{(1)} = -\frac{\lambda N_c}{36\pi^5} \int d^5x \left[ \frac{4h(z)^4 \sqrt{1 - \sqrt[3]{1+z^2}\Phi'(z)^2}}{\sqrt[3]{1+z^2}} + \frac{(h'(z))^2 (1+z^2)}{\sqrt{1 - \sqrt[3]{1+z^2}\Phi'(z)^2}} \right] \tag{15}$$

and the Chern–Simons action contributes as

$$S_{\text{CS}} = \frac{3N_c}{\pi^2} \int h(z)^2 h'(z) \hat{A} \wedge dz \wedge dx_1 \wedge dx_2 \wedge dx_3 \tag{16}$$

as well as a boundary term

$$S_{\text{CS,bdry}} = \frac{3N_c}{4\pi^2} \int_{\text{bdry}} h(\pm\infty)^3 \hat{A} \wedge dx_1 \wedge dx_2 \wedge dx_3 \tag{17}$$

which, however, will vanish when it is evaluated on the solution in our case.

### 2.3. The Single-Layer Solution

In order to have explicit parity invariance, we assume that $h(z) = -h(-z)$. Following [47], we assume that the field $h$ has a discontinuity at $z = 0$, denoted by the blob in Figure 1 (left), and approaches different constant values as $z \to 0$ either from above or from below. As we mentioned above, the discontinuity is required to have a non-vanishing baryon density. Defining the bulk charge density as

$$\rho(z, x^\mu) = -\frac{\delta S}{\delta \, \partial_z \hat{A}_t(z, x^\mu)} \tag{18}$$

the equation of motion for $\hat{A}$ implies

$$\rho'(z) = \frac{3N_c}{\pi^2} h(z)^2 h'(z) \,. \tag{19}$$

The continuous and symmetric solution is given by

$$\rho(z) = \begin{cases} \rho_0 + \frac{N_c}{\pi^2} h(z)^3 \,, & (z < 0) \\ -\rho_0 + \frac{N_c}{\pi^2} h(z)^3 \,, & (z > 0) \end{cases} \tag{20}$$

where

$$\rho_0 = \frac{N_c}{\pi^2} \lim_{z \to 0+} h(z)^3 = -\frac{N_c}{\pi^2} \lim_{z \to 0-} h(z)^3 \tag{21}$$

is the boundary charge density. Notice that, as expected, it is sourced by the discontinuity of $h$. This solution is identified as the single-layer solution. To finalize the construction, we require that $h$ satisfies the equation of motion arising from minimizing the action, except at $z = 0$, where the discontinuity is located.

### 2.4. The Double-Layer Solution

A slightly more general solution than the single-layer solution exists; it is possible that the discontinuity of the $h$ field does not take place at the tip, but at a generic value of the holographic coordinate. The simplest of such solutions, which still respects the symmetry $h(z) = -h(-z)$, is where $h(z)$ vanishes when $-z_c < z < z_c$ so that the discontinuity is located at $z = \pm z_c$; see

Figure 1 (right). Similar solutions were considered in [17] for point-like instantons. In this case, the solution for the bulk charge density is given by

$$
\rho(z) = \begin{cases} \rho_0 + \frac{N_c}{\pi^2} h(z)^3 \,, & (z < -z_c) \\ -\rho_0 + \frac{N_c}{\pi^2} h(z)^3 \,, & (z > z_c) \\ 0 \,, & (-z_c < z < z_c) \end{cases}
\tag{22}
$$

where

$$
\rho_0 = \frac{N_c}{\pi^2} \lim_{z \to z_c+} h(z)^3 = -\frac{N_c}{\pi^2} \lim_{z \to (-z_c)-} h(z)^3 \,.
\tag{23}
$$

This solution is identified as the double-layer solution.

*2.5. Legendre Transform to Canonical Ensemble*

To determine the phase diagram, one needs to determine the free energy, the energy densities, and the grand potential for the different phases. Thus, we first need to compute the free energy of the baryonic phase. For this purpose we first evaluate the on-shell action, as this is related by the holographic dictionary to the four-dimensional free energy density of the field theory (neglecting any singular contributions due to the discontinuity of $h$; see Appendix A). Then, we minimize the action for $h$ to determine the location of the discontinuity.

It is convenient to work at fixed baryonic charge rather than chemical potential. To this end, we perform a Legendre transformation for the action (7):

$$
\widetilde{S} = S + \int \frac{d}{dz} \left( \hat{A}_t \rho \right) dz
\tag{24}
$$

The Legendre transform is introduced only for convenience; in our setup, the computation is simpler in the canonical ensemble, where the Abelian gauge field is not dynamical. One could also work in the grand canonical ensemble. For convenience, we rescale $\rho$ as $\rho \to \frac{4\lambda^4}{531441\pi^6}\rho \equiv \hat{\rho}$. Expanding to first nontrivial order in $h(z)$ and $h'(z)$, and using Equation (18), we can solve for $\Phi'(z)$:

$$
\Phi'(z) = -\frac{\hat{\rho}}{R(z,\hat{\rho})(1+z^2)N_c} - \frac{2187\hat{\rho}\left(-4h(z)^4(1+z^2)^{\frac{2}{3}} + h'(z)^2(1+z^2)^2 R(z,\hat{\rho})^2\right)}{8\lambda^2 N_c (1+z^2)^{\frac{8}{3}} R(z,\hat{\rho})^3} \,,
\tag{25}
$$

where we define

$$
R(z,\hat{\rho}) = \sqrt{1 + \frac{\hat{\rho}^2}{(1+z^2)^{5/3} N_c^2}} \,.
\tag{26}
$$

Then, the Legendre transformed action is

$$
\widetilde{S} = -N_c \int d^5x \left[ \frac{2\lambda^3(1+z^2)^{\frac{2}{3}} R(z,\hat{\rho})}{19683\pi^5} + \right.
$$
$$
\left. + \frac{\lambda\left(4h(z)^4(1+z^2)^{\frac{2}{3}} + h'(z)^2\left((1+z^2)^2 R(z,\hat{\rho})^2\right)\right)}{36\pi^5((1+z^2)R(z,\hat{\rho}))} \right].
\tag{27}
$$

Now we can find the equation of motion for $h(z)$ and solve it. For this purpose, we need to find the appropriate asymptotics of the field $h$ at the boundary:

$$
h \simeq \frac{h_1}{z},
\tag{28}
$$

with $h_1$ remaining as a free parameter.

## 3. Homogeneous Nuclear Matter in the V-QCD Model

V-QCD is bottom-up holographic model that contains both glue and flavor sectors. The glue sector is given by the improved holographic QCD framework [81,82] in which a dilaton field and the potential depending on it are used to implement the essential features of the related QCD sector, i.e., asymptotic freedom, and confinement to deconfinement phase transition. The flavor sector arises from a pair of dynamical space filling flavor branes [83,84]. In V-QCD, the full back-reaction of the flavor branes is taken into account via the Veneziano limit [85], in which both $N_c$ and $N_f$ are large but their ratio is kept $\mathcal{O}(1)$, as it is in real QCD [76]. In the V-QCD flavor sector, a tachyon field is used to realize the breaking/restoration of the chiral symmetry. In both sectors, the model parameters are also fixed by considering perturbative QCD results (running of coupling constant and quark mass) at weak coupling [76,81,82] by requiring qualitative agreement with QCD (e.g., confinement and discrete spectrum) at strong coupling [86], and by fitting to QCD data (e.g., meson and glueball masses and the equation of state at finite temperature) [6,31,87–89]. For a more complete review about the construction of the V-QCD model, the fit to fix the potentials, and comparison with the data, we refer the reader to [19]. In this article, we use one of the models defined in [6] (potentials 7a). This also means that all quark masses are set to zero. The parameter $b$ appearing in the Chern–Simons action [11] is set to $b = 10$.

There are two possible geometries in V-QCD: a horizon-less geometry ending in a "good" kind of singularity [90] (dual to a chirally broken confined phase) and a geometry of a charged "planar" black hole [91,92] (dual to a chirally symmetric deconfined phase). In this article, we focus on the former geometry, which is the relevant geometry for cold and dense nuclear matter. This phase also includes chiral symmetry breaking, which is induced by the condensate of a scalar field $\tau$ (the "tachyon") in the bulk.

In order to discuss nuclear matter, we will employ here an approach that is essentially the same as the homogeneous approach introduced for the WSS model above [11].

This approach has been improved by combining the predictions of V-QCD with other models [93–96]. The resultant equation of states have been widely investigated. It was shown that the resultant equations of state are feasible in the sense of being consistent with neutron star observations [93,94,96–100]. They were also used in phenomenological applications such as modeling spinning neutrons stars [98] and neutron star merger simulations [93,99,100].

In the first two subsections below, we outline the implementation of homogeneous Ansatz in V-QCD and discuss the single-layer solution. For more details, we refer to [19]. In the last two subsections, we present the generalization to double-layer solution and investigate the possibility of a transition from the single-layer to a double-layer configuration.

### 3.1. The Homogeneous Ansatz

For V-QCD, we use the action with finite baryon density, which can be written as

$$S_{V-QCD} = S_{glue} + S_{DBI} + S_{\text{CS}}. \tag{29}$$

The explicit expression for the action can be found in [11]. The renormalization group flow of QCD is modeled through a nontrivial evolution of the geometry between the weak coupling (ultraviolet, UV) and strong coupling (infrared, IR) regions. We will be using here the conformal coordinate $r$ in the holographic direction [81,82] for which the UV boundary is located at $r = 0$ while the IR singularity is at $r = \infty$. As in the case of the WSS model above, we separate the gauge field into non-Abelian and Abelian components:

$$\mathcal{A}_{L/R} = A_{L/R} + \hat{A}_{L/R} \,. \tag{30}$$

Here, the left- and right-handed fields arise from $D4$ and $\overline{D4}$ branes, respectively [83,84]. Similarly, as in the case of the WSS model above, we turn on the temporal component of the vectorial Abelian gauge field

$$\hat{A}_L = \hat{A}_R = \mathbb{I}_{N_f \times N_f} \Phi(r) dt . \tag{31}$$

Then, on top this background, the non-Abelian baryonic terms are treated as a perturbation. We expand the DBI action up to a first nontrivial order in the non-Abelian fields (quadratic in the field strengths $F_{(L/R)}$).

After the expansion, we insert the homogeneous Ansatz for non-Abelian gauge field, i.e.,

$$A_L^i = -A_R^i = h(r)\sigma^i \tag{32}$$

where $h(r)$ is a smooth function and $\sigma^i$ are Pauli matrices introducing the non-trivial flavor dependence $SU(2)$. As result, the action for the flavor sector is written as

$$S_h = S_{DBI}^{(0)} + S_{DBI}^{(1)} + S_{CS} \tag{33}$$

where $S_{DBI}^{(0)}$ is the DBI action in the absence of solitons, $S_{DBI}^{(1)}$ is the expansion of the DBI action with homogeneous Ansatz at the second order, and $S_{CS}$ is the Chern–Simons term with the homogeneous Ansatz (the explicit expressions are given in [11]).

### 3.2. The Single-Layer Solution

The solution for the bulk charge density is found by considering the $\Phi$ equation of motion [11]

$$\rho' = -\frac{d}{dr}\frac{\delta S_h}{\delta \Phi'} = -\frac{\delta S_h}{\delta \Phi} = \frac{2N_c}{\pi^2}\frac{d}{dr}\left[e^{-b\tau^2}h^3(1 - 2b\tau^2)\right], \tag{34}$$

where $b$ is a parameter in the Chern–Simons term, $\rho$ is the bulk charge density, and $\tau$ is the tachyon field. However, the solution for $\rho$ implied by this equation vanishes both in the UV and in the IR. That is to say, diverging tachyons in the IR set the solution to zero via the exponential factor and the boundary condition for $h$ in UV requires it to vanish (since there is no external baryon source). Therefore, as was the case in the WSS model above, the baryon density is zero, unless we impose an abrupt discontinuity in the field $h$.

Motivated by these considerations, we write the "single-layer" solution for V-QCD as [11]

$$\rho = \begin{cases} \rho_0 + \frac{2N_c}{\pi^2}e^{-b\tau^2}h^3(1 - 2b\tau^2), & (r < r_c) \\ \frac{2N_c}{\pi^2}e^{-b\tau^2}h^3(1 - 2b\tau^2), & (r > r_c) \end{cases} \tag{35}$$

where $\rho_0$ is boundary baryon charge density (the physical density) and $r_c$ is the location of the discontinuity. The explicit expression for $\rho_0$ is

$$\rho_0 = \frac{2N_c}{\pi^2}e^{-b\tau(r_c)^2}(1 - 2b\tau^2(r_c))\text{Disc}(h^3(r_c)), \tag{36}$$

where we use the notation $\text{Disc}(g(r_c)) \equiv \lim_{\epsilon \to 0+}(g(r + \epsilon) - g(r - \epsilon))$.

For future convenience, we briefly discuss the asymptotics of the field $h$. In the UV, $h$ has the asymptotics typical for gauge fields:

$$h \simeq h_1 + h_2 r^2 . \tag{37}$$

We require that non-Abelian sources vanish; therefore, $h_1 = 0$, but $h_2$ remains as a free parameter (which also determines $r_c$ for given $\rho_0$; see Appendix A.2). Following [11], we set $h(r) = 0$ for $r > r_c$.

### 3.3. The Double-Layer Solution

In this subsection, we generalize the single-layer solution for baryon field $h$ to have two discontinuities, i.e.,

$$\rho = \begin{cases} \rho_{01} + \frac{2N_c}{\pi^2}e^{-b\tau^2}h^3(1 - 2b\tau^2), & (r < r_{c1}) \\ \rho_{02} + \frac{2N_c}{\pi^2}e^{-b\tau^2}h^3(1 - 2b\tau^2), & (r_{c1} < r < r_{c2}) \\ \frac{2N_c}{\pi^2}e^{-b\tau^2}h^3(1 - 2b\tau^2), & (r > r_{c2}) \end{cases} \tag{38}$$

which we will be called the double-layer solution. There is also a continuity condition on $\rho$ that must be satisfied, which is given as

$$\rho_{02} = -\frac{2N_c}{\pi^2}(1 - 2b\tau^2)e^{-b\tau^2}\mathrm{Disc}\, h^3|_{r=r_{c2}}\,, \qquad \rho_{01} - \rho_{02} = -\frac{2N_c}{\pi^2}(1 - 2b\tau^2)e^{-b\tau^2}\mathrm{Disc}\, h^3|_{r=r_{c1}}\,. \tag{39}$$

Therefore, summing the equalities above, we identify the boundary baryon charge density $\rho_0$ as $\rho_{01}$:

$$\rho_0 = \rho(r=0) = \rho_{01} = -\frac{2N_c}{\pi^2}\sum_{i=1}^{2}(1 - 2b\tau^2)e^{-b\tau^2}\mathrm{Disc}\, h^3|_{r=r_{ci}}\,. \tag{40}$$

We stress, however, that even though we call this solution by the same name as the double-layer solution for the WSS model, these solutions are quite different. In particular, the double-layer V-QCD solution has discontinuities at two values of the holographic coordinate, whereas the WSS solution only has discontinuities at a single value. Actually, the single-layer solution of the V-QCD model is closer to the double-layer solution of the WSS model than the double-layer solution of the V-QCD model. We will discuss this difference in more detail below.

The double-layer solution depends on four parameters at fixed $\rho_0$; there is one additional parameter from the location of the extra discontinuity with respect to the single-layer solution, and as the solution for $h$ in the second interval $r_{c1} < r < r_{c2}$ is independent of the solution in the first interval, there are two additional integration constants from the solution of $h$. Finally, the generalization to triple-layer solution or even to a solution with a higher number of flavors is straightforward. One only needs to modify the piecewise solution for the charge density $\rho$ with addition of new intervals. This will introduce three new parameters for each interval.

*3.4. Legendre Transform to Canonical Ensemble*

As in the analysis of the WSS model above, it is convenient to work in the canonical ensemble. The Legendre transformed action for V-QCD becomes [11]

$$\widetilde{S}_h = -\int d^5 x V_\rho G \sqrt{1 + \frac{\rho^2}{(V_\rho w e^{-2A})^2}} \left[1 + \frac{6w^2 e^{-4A} h^4 + 6\kappa\tau^2 e^{-2A} h^2}{1 + \rho^2(V_\rho w e^{-2A})^{-2}} + \frac{3}{2}\frac{w^2 e^{-4A} f(h')^2}{G^2}\right]. \tag{41}$$

**4. Results**

*4.1. Second-Order Transition in the Witten–Sakai–Sugimoto Model*

We start by analyzing the configurations in the WSS model. We set $\lambda = 16.63$ [46]) and analyze the solutions numerically (see Appendix A). As a function of the chemical potential, we find three phases (see Figure 2, where we show the grand potential and the baryon charge density as a function of the chemical potential):

1.　Vacuum for $\mu < \mu_c$ with $\mu_c \simeq 0.205$;
2.　Single-layer phase for $\mu_c < \mu < \mu_l$ with $\mu_l \simeq 0.342$;
3.　Double-layer phase for $\mu > \mu_l$.

The phase transition at $\mu = \mu_c$ ($\mu = \mu_l$) is of the first (second) order. Here the second-order transition (at the higher value of the chemical potential, $\mu = \mu_l$), is identified as the popcorn transition. Notice that in the approach of [50], which used a different variation of the homogeneous approach, both the vacuum-to-nuclear and popcorn transitions were of the first order. Even though we are not attempting a serious comparison to QCD data, we note that by setting $M_{KK} = 949$ MeV as determined by the mass of the $\rho$ meson [46], we have (for the quark chemical potential) $\mu_c \simeq 195$ MeV and $\mu_l \simeq 325$ MeV, i.e., numbers in the correct ballpark. We note that $\mu_l/\mu_c \simeq 1.67$. Denoting the density of the single-layer configuration at $\mu = \mu_c$ as $\rho_c$ (i.e., the analogue of the saturation density), the density $\rho_l$ at the second-order transition satisfies $\rho_l/\rho_c \simeq 3.4$.

Here, we are mostly interested in the second-order transition from the single- to double-layer phase. We show the relevant configurations in Figure 3 for a choice of densities $\rho_0$ around the critical value $\rho_l \simeq 2.52 \times 10^{-4}$. Recall that the single-layer configuration is unique for fixed $\rho_0$, whereas the double-layer configuration also depends on $z_c$. We show here the double-layer profiles, which minimize the free energy. They are separate from the single-layer configuration only for $\rho_0 > \rho_l$ (the three highest values in the figure), where they have lower free energies than the single-layer solutions. Interestingly, the single- and double-layer solutions at the same $\rho_0$ are close; the functions $h(z)$ deviate by at most a few percentage points in the region $z > z_c$. The deviations for $\rho(z)$ are slightly higher, and the single-layer solution can be viewed as a smoothed out version of the double-layer solution. That is, even if we were not considering the double-layer solutions explicitly, their presence could be guessed from the single-layer solutions. In both cases, deviation is largest close to $z_c$. We also remark that the single-layer profiles $h(z)$ appear to be qualitatively similar to the solutions found in the approach of [50] (see Figure 4 in this reference), up to a shift by a constant.

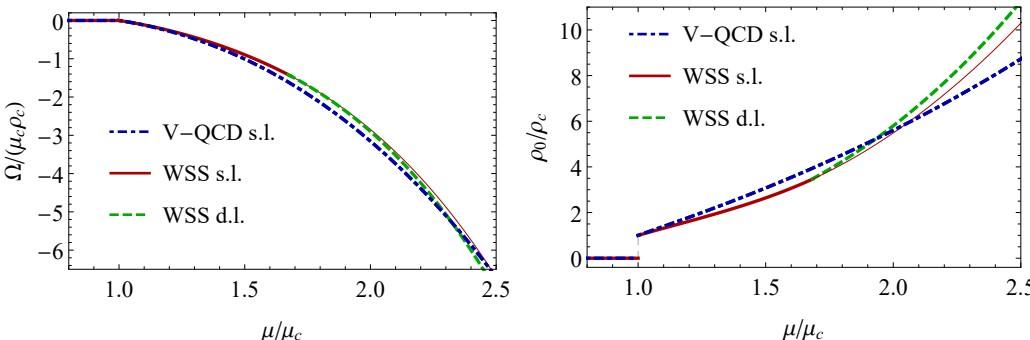

**Figure 2.** The normalized grand potential $\Omega/(\mu_c\rho_c)$ (**left**) and the normalized charge density $\rho_0/\rho_c$ (**right**). The solid red, dashed green, and dot-dashed blue curves are the results for the single-layer configuration in the WSS model, double-layer configuration in the WSS model, and the V-QCD model, respectively. The vacuum-to-nuclear-matter transitions occur at $\mu/\mu_c = 1$.

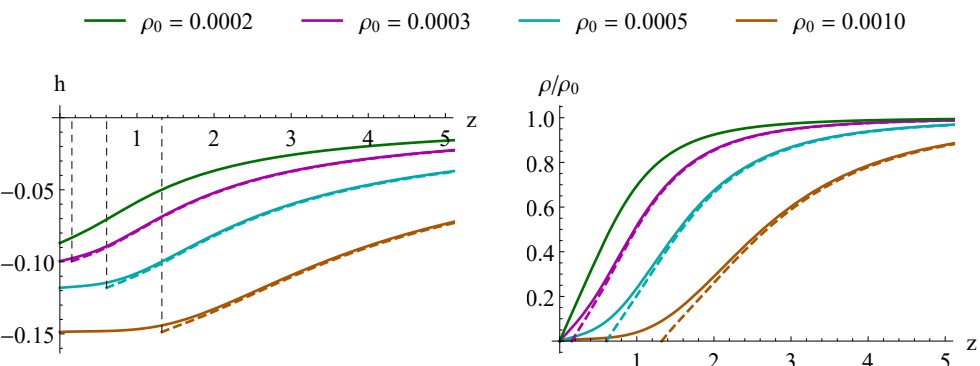

**Figure 3.** The profile of the gauge field $h(z)$ (**left**) and the bulk charge density $\rho(z)$ (**right**) for the single-layer (solid curves) and double-layer (dashed curves) configurations for various values of the charge density. The vertical dashed lines in the left hand plot denote the discontinuities of the double-layer solutions at $z = z_c$.

### 4.2. Analysis of Configurations in V-QCD

We construct the double-layer and single-layer solutions using the procedure outlined in Appendix A.2. The essence of the procedure is the minimization of the free energy density at fixed $\rho_0$, depending on the free parameters. In the case of the single layer, there is only one parameter, $r_c$, or equivalently, $h_2$, and it is straightforward to solve the equation of state in this case. The results of this minimization procedure (the grand potential and the baryon charge density as a function of chemical potential) for the single-layer solution are shown in Figure 2 with the blue dot-dashed curves in the plots.

For the double-layer solution, there are four parameters that make the numerical minimization procedure challenging in contrast to the single-layer solution. Therefore, while we perform minimization of the single-layer solution for a large domain of $\rho_0$ values, we investigate presence of the lower free energy density of the double-layer solution only for solutions obtained by gluing together single-layer solutions for some representative values of $\rho_0$, changing from 0.8 to 2.5.

Denoting $\Delta h_i = \text{Disc } h(r_{ci})$, we investigate three qualitatively different configurations. We consider $\Delta h_1 < 0$ and $\Delta h_1 > 0$ for a double-layer solution and $\Delta h_1 > 0$, $\Delta h_2 > 0$ for a triple-layer solution. For boundary baryon number charge, we consider the values of $\rho_0 = 0.5$, $\rho_0 = 0.8$ and $\rho_0 = 2.5$, which correspond to $\mu/\mu_c = 1.65$, $\mu/\mu_c = 2.04$ and $\mu/\mu_c = 3.57$ for the thermodynamics determined by the single-layer solution, respectively. While the first choice roughly corresponds to chemical potential values in which double-layer solutions in WSS become dominant (as it is seen from Figure 4 below), the other two choices are even larger than that.

In Figure 4, the results for the three representative case are shown. The baryon field profile $h(r)$ and corresponding baryon number densities $\rho_0(r)$ in the bulk are shown in the first and second column, respectively. In each plot, the single-layer solution minimizing the free energy is shown with gray dashed curves whose parameters are given in the first row of Table 1. The red, blue, and green solid curves show $\Delta h_1 < 0$ and $\Delta h_1 > 0$ double-layer and ($\Delta h_1 > 0$, $\Delta h_2 > 0$) triple-layer solutions. The parameters $r_{ci}$, $h_{2i}$, $\rho_{0i}$, where $h_{2i}$ are the asymptotic constants $h_2$ for the single-layer solutions that were glued together to obtain the multilayer solutions, and the corresponding free energy densities $f$ are shown in Table 1. The locations of the discontinuities and $\rho_{0i}$ are also shown in the figures with the blobs.

We were able to find double-layer solutions that have lower free energy than the single-layer solution at a fixed charge density for the cases of $\Delta h_1 > 0$ i.e., the second row of Figure 4. However, we were not able to find double-layer solutions with $\Delta h_1 < 0$ that would have lower free energy than the single-layer solution (configurations in the first row of the figure). Notice that having solutions with $\Delta h_1 > 0$ means that contributions to the total charge from the two discontinuities have opposite signs. This means that in the instanton picture, the discontinuities must arise from smearing instantons with opposite charges. This suggests that proton–antiproton pairs are created, which should be forbidden due to the large energy required for such a pair creation. Therefore, the configuration of the first row is not physically sound. We suspect that it appears because the homogeneous approximation works poorly with configurations with discontinuities at several values of the holographic coordinate. We also show the example of a triple-layer configuration with $\Delta h_1 > 0$ and $\Delta h_2 > 0$ on the third row of the plot.

**Table 1.** The values of $\{r_c, h_2, \rho_0, f\}$ for the single-layer configuration (first row) and $\{r_{ci}, h_{2i}, \rho_{0i}, f\}$ for the multi-layer configurations (second–fourth rows) shown in Figure 4.

| $r_c$ | $h_2$ | $\rho_0$ | f |
|---|---|---|---|
| 0.570 | 2.991 | 0.80 | 1.745 |
| $\{0.483, 0.539\}$ | $\{3.90, 3.10\}$ | $\{0.80, 0.59\}$ | 2.003 |
| $\{0.487, 0.525\}$ | $\{2.90, 4.10\}$ | $\{0.80, 0.72\}$ | 1.626 |
| $\{0.476, 0.498, 0.533\}$ | $\{2.90, 3.20, 3.90\}$ | $\{0.80, 0.74, 0.64\}$ | 1.642 |

### 4.3. Speed of Sound and Polytropic Index

We now study the physical implications of the phase transition. To this end, we plot the speed of sound and the polytropic index $\gamma = d \log p / d \log \epsilon$ for the WSS and V-QCD models in Figure 5. In these plots, the chemical potential was normalized using the value at the vacuum-to-nuclear-matter transition.

In both models, the speed of sound is below the value $c_s^2 = 1/3$ of conformal theories right above the transition to nuclear matter. When $\mu$ increases, however, the speed of sound

crosses this value and reaches values well above it [11]. The speed of sound has a maximum in both model. Even though the location of the maximum is different between the models, the maximal values are rather close; the maximum of $c_s^2$ is 0.463 (at $\mu/\mu_c = 1.355$) for the WSS model and 0.504 (at $\mu/\mu_c = 2.246$) for V-QCD. Eventually, at higher densities, the speed of sounds decreases to values closer to the conformal value. This is clearer in the WSS than in the V-QCD model. In the WSS model, where the popcorn transition from a single- to a double-layer configuration is found, the speed of sound drops to a roughly constant value, which closely agrees with the conformal value in the double-layer phase; the speed of sound squared is about one percent higher than the conformal value $1/3$.

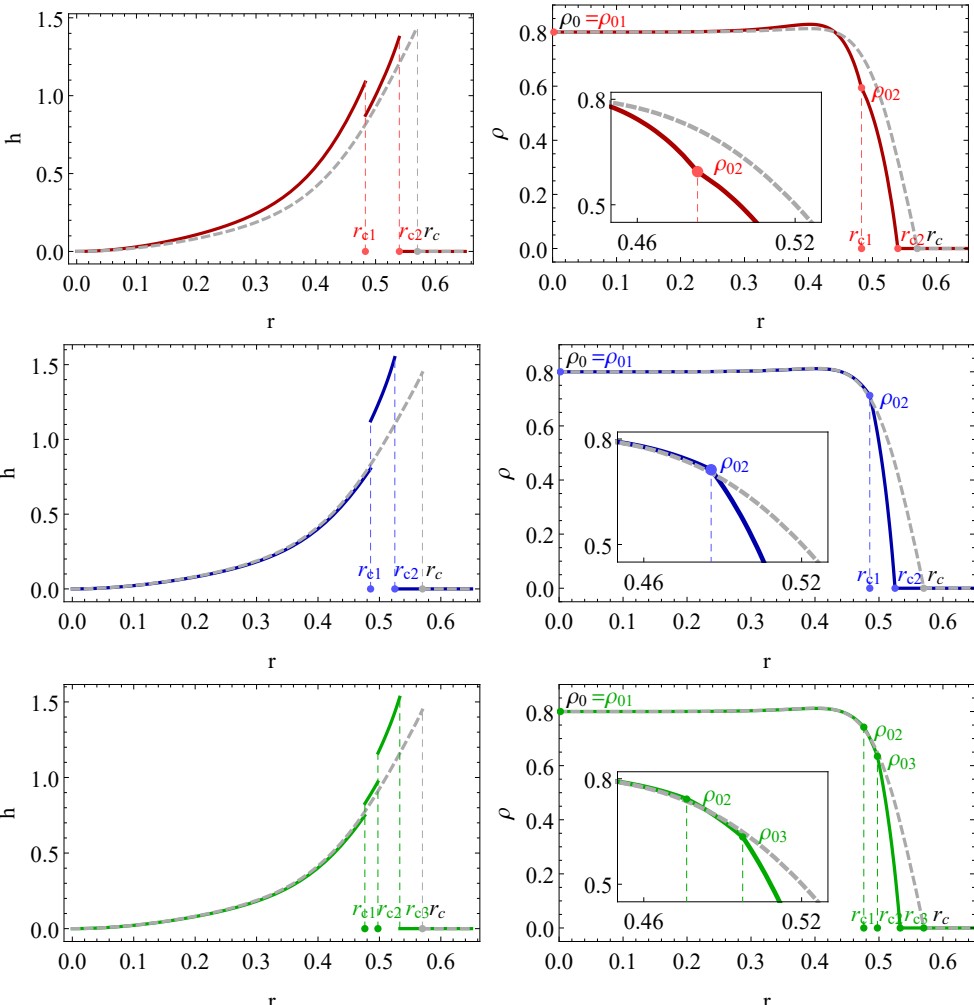

**Figure 4.** The profile of the gauge field $h(r)$ (**left**) and the bulk charge density $\rho(r)$ (**right**) for a double-layer configuration with $\Delta h_1 < 0$ (first row), double-layer configuration with $\Delta h_1 > 0$ (second row), and triple-layer configuration with $\Delta h_1 > 0$ and $\Delta h_2 > 0$ (third row). The single-layer configuration with the same boundary charge density $\rho_0 = 0.8$ is shown with the gray dashed curve in each plot. The parameters $r_{ci}$ and $\rho_{0i}$ that characterize the multi-layer configurations are shown by blobs. The values of $r_{ci}$, $h_{2i}$, $\rho_{0i}$, and $f$ are given in Table 1.

Similar results are found for the polytropic index $\gamma$ in the right-hand plot of Figure 5. In both models, $\gamma$ decreases with $\mu$ in the (single-layer) nuclear matter phase. This decrease is fast in the sense that $\gamma$ drops below the value of $\gamma = 1.75$, which was used as a criterion to separate nuclear matter from quark matter in [67,68], where equations of state obtained as interpolations between known results from nuclear theory at low density and perturbation theory at high density were considered. For $\mu/\mu_c > 1.5$, the results from both model are below this value. At the popcorn transition of the WSS model, $\gamma$ drops to a value close to one.

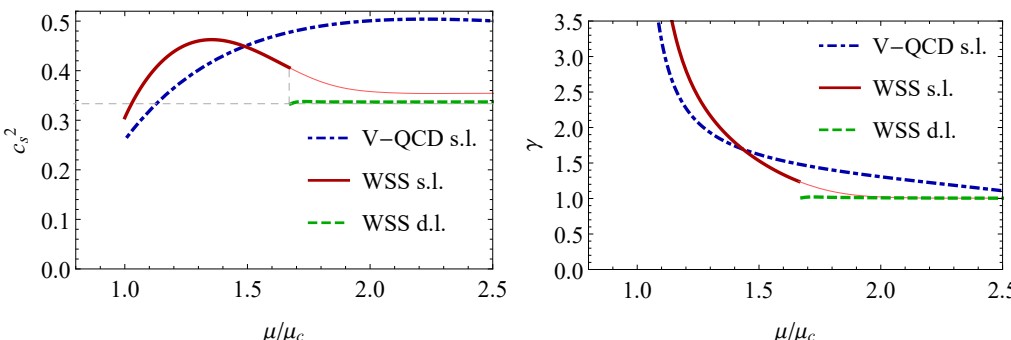

**Figure 5.** The speed of sound (**left**) and the polytropic index $\gamma = d\log p/d\log \epsilon$ (**right**). The solid red, dashed green, and dot-dashed blue curves are the results for the single-layer configuration in the WSS model, double-layer configuration in the WSS model, and the V-QCD model, respectively.

Our findings indicate that the homogeneous holographic nuclear matter behaves approximately conformally at high densities, i.e., at densities well above the nuclear saturation density (see also [101]). This is particularly clear for the WSS model, which becomes approximately conformal at the popcorn transition. These findings are consistent with earlier studies of homogeneous nuclear matter in the WSS (see, e.g., [102]) and the V-QCD (see, e.g., [94]) models. They also agree with the results found in the effective theory approach of [62,64]. This agreement is strikingly good for the WSS model, where the results both for the speed of sound (see [64]) and for the polytropic index (see [65]) have been computed. For example, our results for the maximal value of the speed of sound (our value is $c_{s,max} \approx 0.68$) and the density at the popcorn transition (we found $n_l/n_c \approx 3.4$) agree rather well with those of these references; our value for the speed of sound (transition density) is a bit below (above) the values of the effective theory approach.

We also remark that the non-monotonic behavior of the speed of sound in the WSS model qualitatively agrees with that found in the point-like instanton gas approach in [15], albeit with a different embedding for the $D8$ branes. The maximal value found in this reference is also close to the maximal value obtained here. This agreement is interesting, as the results were obtained in a completely different approach, which is expected to be reliable at lower densities. Moreover we compare our results to the different approach of homogeneous nuclear matter derived in [50] in Appendix B, and mostly find qualitative agreement.

## 5. Conclusions

In this article, we analyzed nuclear matter using a homogeneous approach in two different holographic models: the top-down WSS model and the bottom-up V-QCD model. We focused on two topics: popcorn transitions, where the layer structure of the nuclear matter changes in the bulk, and approach to conformal behavior at high densities. We found a second-order popcorn transition in the WSS model and signs of approach to conformality in both holographic setups.

We have several remarks about our results. Firstly, the results in the WSS and V-QCD models appeared to be quite different; in particular, the popcorn transition was only found to take place in the WSS model. However, this is not surprising at all and can be seen to follow from the differences in the geometry and the realization of chiral symmetry breaking between the models, as we now explain. Recall that in the WSS model, the geometry ends at the tip of the cigar in the confined phase as shown in Figure 1, and chiral symmetry breaking is realized by the joining of the two branches of flavor branes at the tip. In the V-QCD picture, there is no cigar structure, and chiral symmetry breaking arises from a condensate of a bulk scalar field. In the WSS model, nuclear matter at low densities is seen to arise from instantons located at the tip, and it is not possible to assign such instantons to be left- or right-handed. In V-QCD, however, nuclear matter is stabilized at a nontrivial value of the holographic coordinate due to interaction with the bulk scalar field [11] and,

by definition, always contains left- and right-handed components. Therefore, in V-QCD, separate configurations analogous to the single- and double-layer configurations of WSS in Figure 1 do not exist. The configurations of this figure map have the same configuration as in V-QCD, which is what we called the single-layer configuration. The double-layer configuration in V-QCD defined in (38) would map to a more complicated configuration in the WSS model, where discontinuities of the $h$ field are found at two distinct values of $z$.

We found that the results for the equation of state near the popcorn transition of the WSS model closely resemble those obtained by the framework of [62,64], where effective theory was used to analyze the transition of the skyrmion crystal to a crystal of half-skyrmions. This suggests that the transition in the holographic model should be identified with the topology changing transition where half-skyrmions appear (we thank N. Kovensky and A. Schmitt for correspondence on this question). It is, however, difficult to say anything definite about this because the holographic approach that we used does not contain individual instantons. Moreover, in [50], it was argued that the topology changing transition should not be identified as the transition between the single- and double-layer solutions, but should take place between solutions of qualitatively different behavior within the single-layer solution. Another point is that chiral symmetry should be restored globally at the topology changing transition (meaning that the averages of the condensate over large regions should vanish). This, however, will not happen for any of the configurations in the WSS approach because the $D8$ brane action is treated in the probe approximation, and the embedding of the brane is independent of the density. Nevertheless, we remark that, as seen from the expressions for the single- and double-layer configurations in (20) and in (22), the bulk charge density has support near the tip of the cigar only for the single-layer configuration, where the flavor branes join, breaking chiral symmetry. Therefore, the double-layer configuration can also exist in chirally symmetric backgrounds. Examples of such chirally symmetric double-layer configurations were indeed found in [17] (the chirally symmetric quarkyonic matter phase of this reference).

Finally, we demonstrated that the homogeneous nuclear matter becomes approximately conformal at high densities, i.e., a few times above the nuclear saturation density. That is, the values of the speed of sound lie close to the value $c_s^2 = 1/3$ of conformal theories, and similarly $\gamma$ values lie close to the value $\gamma = 1$. In particular, the polytropic index reached values well below the value $\gamma = 1.75$ both in the V-QCD model and in the WSS model, which has been used to classify equations of state for nuclear and quark matter in the approach of [67,68]. That is, part of the single-layer phase and all of the double-layer phase would be classified as quark matter in this approach. This appears consistent with the interpretation that the double-layer phase is smoothly connected to quark matter [69]. In the V-QCD setup, however, there is a separate strong first-order phase transition from nuclear to quark matter at higher densities [11,94]. In the WSS model, there is a separate quark matter phase as well, but in this case, the transition is weak and even continuity between the phases is a possibility [10].

**Author Contributions:** Investigation, J.C.R., T.D. and M.J. All authors have read and agreed to the published version of the manuscript.

**Funding:** J.C.R. and M.J. have been supported by an appointment to the JRG Program at the APCTP through the Science and Technology Promotion Fund and Lottery Fund of the Korean Government. J.C.R. and M.J. have also been supported by the Korean Local Governments—Gyeong-sangbuk-do Province and Pohang City—and by the National Research Foundation of Korea (NRF) funded by the Korean government (MSIT) (grant number 2021R1A2C1010834). T.D. acknowledges the support of the Narodowe Centrum Nauki (NCN) Sonata Bis Grant No. 2019/34/E/ST3/00405.

**Institutional Review Board Statement:** Not applicable.

**Informed Consent Statement:** Not applicable.

**Data Availability Statement:** Not applicable.

**Acknowledgments:** We thank Mannque Rho for the invitation to contribute to the special issue "Symmetries and Ultra Dense Matter in Compact Stars" in Symmetry. We also thank Elias Kiritsis, Nicolas Kovensky, Yong-Liang Ma, and Andreas Schmitt for discussions and correspondence. This work benefited from discussions during the APCTP focus program "QCD and gauge/gravity duality".

**Conflicts of Interest:** The authors declare no conflict of interest.

## Appendix A. Numerical Details

*Appendix A.1. Constructing the Solution in the Witten–Sakai–Sugimoto Setup*

Here, we summarize the basic steps we followed to find the free energy and the equation of state for the case of the simple profile for the charge density (20):

1.  We derive from the action (27) the equation of motion for $h(z)$. After plugging the baryon charge density $\rho$ and fixing $N_c \to 3$ and $\lambda \to 16.63$, the only free parameter is the boundary charge density $\rho_0$. Then, we can simply solve the equation for $h(z)$ for fixed $\rho_0$ from the UV boundary (we still need to fix $h_1$);

2.  We fix the value of $h_1$ by solving for $h$ for a given fixed $\rho_0$ and choose a value of $h_1$ such that $\rho(h) = 0$ at $z = 0$. After this, we can determine the bulk charge density $\rho$ profile by considering (20);

3.  The free energy density is given by explicit integration of (27) from zero to a large cut-off. At this step, we (re)normalize the free energy by subtracting $\widetilde{S}$ in the absence of baryons from the original $\widetilde{S}$. Notice that this prescription means that singular contributions at the discontinuity of $h$ are neglected. When $h$ is discontinuous, $h'(z)^2$ is, in principle, proportional to the delta function squared, but such contributions are not taken into account;

4.  From the tabulated data $\{\rho_0, F\}$, we can construct $F(\rho_0)$ and find at which value of $\rho$ the transition to nuclear matter happens. The corresponding chemical potential and grand potential can be obtained via $\mu = dF/d\rho_0$ and $\Omega = F - \rho_0\mu = -p$.

For the case of the more general solution (22), we need to find the value $z_c$ where the charge density vanishes; then, the procedure to find the energy as a function of $\rho_0$ is analogous to the single-layer solution above. However, one difference with respect to the previous single-layer solution is that the value of $h_1$ that minimizes the energy changes for densities larger than a critical value.

From the comparison of the free energy, we can see that there is a second-order phase transition at this critical density $\rho_c$ from the single-layer solution to the double-layer solution.

*Appendix A.2. Constructing the Solution in the V-QCD Setup*

In this subsection, we summarize and outline the calculation of free energy density and the minimization procedure:

1.  We work in the probe limit. We first construct the thermal gas background solution for the geometry [76] in the absence of the baryons;

2.  Then, from (41), we derive equations of motion for $h$. After plugging background fields and baryon charge density $\rho$, the only free parameter is the boundary charge density $\rho_0$. Thus, we can simply solve the equation of motion for $h$ for fixed $\rho_0$ by UV boundary;

3.  After solving for $h$ for a given fixed $\rho_0$ and choosing $h_2$, we can determine the bulk charge density $\rho$ profile by considering (35). Note that the vanishing point of bulk density profile gives the location of the soliton, i.e., $\rho(r_c) = 0$;

4.  The free energy density is given by explicit integration of (41) from the boundary to the location of the discontinuity. At this step, we also subtract $\widetilde{S}_h$ in the absence of baryons from the original $\widetilde{S}_h$ to (re)normalize the free energy;

5. Now, we can return to our main purpose of minimizing free energy at a fixed $\rho_0$ depending on the free parameter $r_c$ or, equivalently, $h_2$. We can simply perform abovementioned procedure with a loop over $h_2$ values;

6. From the tabulated data, we can construct $F(h_2)$ and minimize it. The corresponding chemical potential and grand potential can be obtained via $\mu = dF/d\rho_0$ and $\Omega = F - \rho_0\mu = -p$.

For the case of the multi-layer configurations, the number of parameters that should be used in the minimization procedure increases, making similar analysis numerically expensive. This is beyond the scope of this project. Therefore, we decided to analyze the situation by considering some representative situations (the details of which are given in the main text in Section 4.2) and searching for solutions with lower $f$ than that of single-layer configurations.

## Appendix B. Comparison to a Different Homogeneous Approach

In this appendix, we compare our results to those obtained by employing the homogeneous approach of [50], where one uses a zero curvature condition before taking the system to be homogeneous in the WSS model. We set the parameter $\Lambda = 8\lambda/(27\pi)$ to the value $\Lambda \approx 1.568$ for consistent comparison with our approach. The results are shown in Figure A1. We see that the maximal value of the speed of sound is higher in the approach of [50] than in the other approaches, and the value of $\mu$ at the popcorn transition is likewise higher than in the approach we used here. For the ratio of transition densities of single-layer nuclear matter, we find $\rho_l/\rho_c \approx 8.0$. Similarly, the value of the polytropic index $\gamma$ is relatively high when using the approach of [50]. This also means that the agreement with the effective theory model of [62,64], which was discussed in the main text, is worse for this approximation.

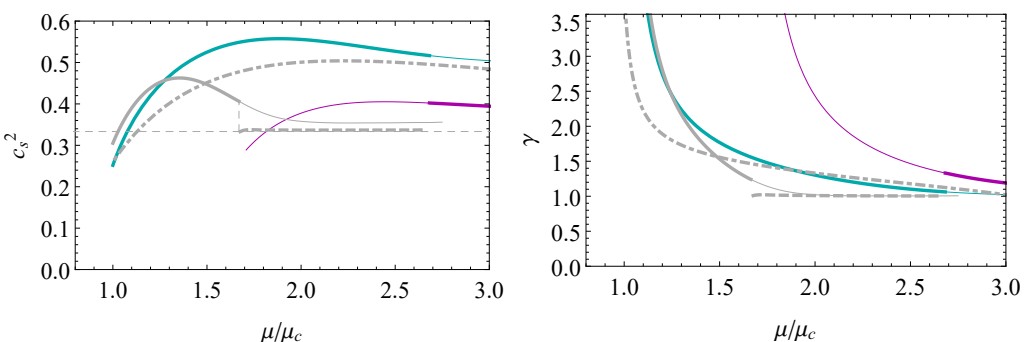

**Figure A1.** The speed of sound (**left**) and the polytropic index $\gamma = d\log p/d\log\epsilon$ (**right**) for single-layer (cyan curves) and double-layer (magenta curves) solutions in the approach of [50]. The gray curves show the WSS and V-QCD results presented in Figure 5.

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
