# Peer review of "Popcorn Transitions and Approach to Conformality in Homogeneous Holographic Nuclear Matter"

_symmetry, doi:10.3390/sym15020331_

Round 1

Reviewer 1 Report

This paper addresses dense nuclear matter relevant to the structure of massive compact stars. Given that baryonic matter at densities exceeding the density $n_0$ of the equilibrium nuclear matter by a factor of 2 or more cannot be accessed quantitatively by QCD, the equation of state (EoS) of dense compact stars remains an uncharted domain of strong interaction physics.  It remains a challenge both to nuclear physics and to astrophysics. Currently there are a large number of approaches developed in the field with an abundance of papers published thereon spurred by gravity-wave anchored observations. But there has been very little progress made in the field.

At present, there are, broadly speaking,  two major avenues that are promising:  one bottom-up (BU) and the other top-down (TD). The former starts with what’s known in low-energy nuclear physics and attempts to approach the pertinent density regime by trial and error, with the help of poorly available experimental data. It’s anchored on effective field theories (EFTs) which are inevitably, by construction, to break down at a few times $n_0$. Going beyond in the standard EFTs is unknown. The latter approach, on the contrary, starts from perturbative QCD at asymptotic density  and attempts to go down to the density regime extrapolated from EFT. How to join it to the EFT regime is not under control. 

It is the gravity-gauge dual holographic approach adopted in this paper that offers the possibility to achieve the joining of the BU and the TD. The TD approach, e.g., the Sakai-Sugimoto model, implements an infinite tower of mesons, thereby going beyond the hidden local symmetry (HLS) approach of low-density theories. Although its ultraviolet completion is not known, it nonetheless goes toward the high density regime of massive stars. The BU approach, with the incorporation of hidden symmetries, e.g., HLS and hidden scale symmetry implementing “genuine dilaton” etc., can join the lower edge of perturbative QCD. In this paper, the authors describe how this matching can be done in a reasonably realistic way. By doing so, they see the emergence of conformal symmetry at increasing density in the speed of sound in the core of massive stars. A quantitatively reliable way to expose the conformal sound speed does not seem to have been achieved yet in the paper but it appears to be going in the right direction. The approach has also the potential to expose other symmetries hidden in QCD, such as for instance Chern-Simons topological structure associated with fractional quantum Hall droplet matter discussed by other contributors to this Special Issue.

I recommend that this paper be accepted for publication. The only request to the authors is that the authors correct various dangling statements that make some of the discussions obscure.

Author Response

We thank the referee for the careful assessment and positive comments. We are also ready to make improvements in the text. Please let us now which parts of the text are unclear or imprecise, and we will try to fix them.

Reviewer 2 Report

In my opinion, the paper is written well enough to recommend it for publishing in the present form.

Author Response

We thank the referee for reviewing the manuscript and for the positive comments.

Reviewer 3 Report

This article conducts a study of nuclear matter at high density by evaluating two string-inspired holographic models of QCD.  As is standard, the 5-dim effective action for the model is constructed with a classical (Abelian) gauge potential for the finite chemical potential.  The 5-d action is minimized as a function of baryon charge density to determine the equation of state.

Overall the study is carried out well.  A few questions should be clarified in order to make it more readable, notation explained and most importantly approximations justified.

[Lines 60-70] It would be helpful to provide context/motivation how the chosen models manifest nuclear degrees of freedom that are known to be relevant in the approach to any phase transition from the low density side.

[Eq 5] The calligraphic F is not defined.

[Above Eq 7] F^2 is used as is common for the Lorentz scalar, but is not defined as such and makes one uncertain about the calligraphic F above.

[Line 133] Is Nf=2 a good approximation given that the critical chemical potential is estimated >2m_s?

[Eq 27 and 41] How is the 5-d dimensionless action related to the 4-d free energy density (a dimension 4 quantity)?  What is achieved by the Legendre transformation?

[Fig 2] What are the units of rho0?  Are they the same as the normalized rho-hat?  Or do they have dimension?

[Line 280] Under what conditions are the 2-layer solutions lower energy than the 1-layer solutions?  Given the unphysical nature of the 2-layer solutions and instability of any other non-minimum-energy solutions, this results calls into question the assumptions—as it should, see next.

[Line 289]  The homogenous approximation is an assumption about the physics of the nuclear matter, so it seems this model produces unphysical solutions.  The question is whether physical (but unstable) solutions exist for values of the model parameters that do not minimize the energy.  It is in any case thought that nuclear matter in this regime is not homogeneous.

[Line 412/413] ‘chose’ -> ‘choose’ and this is a run-on sentence

[Line 418] It would be helpful to show the free energy since it has been constructed
